# Hiding in Plain Sight: Virtually Unrecognizable Memory Phenotype CD8^+^ T cells

**DOI:** 10.3390/ijms21228626

**Published:** 2020-11-16

**Authors:** Daniel Thiele, Nicole L. La Gruta, Angela Nguyen, Tabinda Hussain

**Affiliations:** Department of Biochemistry and Molecular Biology, Biomedicine Discovery Institute, Monash University, Clayton VIC 3800, Australia; daniel.thiele@monash.edu (D.T.); nicole.la.gruta@monash.edu (N.L.L.G.)

**Keywords:** virtual memory T cells, Eomes, IL-15, CD8^+^ T cells

## Abstract

Virtual memory T (T_VM_) cells are a recently described population of conventional CD8^+^ T cells that, in spite of their antigen inexperience, express markers of T cell activation. T_VM_ cells exhibit rapid responsiveness to both antigen-specific and innate stimuli in youth but acquire intrinsic antigen-specific response defects in the elderly. In this article, we review how the identification of T_VM_ cells necessitates a re-evaluation of accepted paradigms for conventional memory T (T_MEM_) cells, the potential for heterogeneity within the T_VM_ population, and the defining characteristics of T_VM_ cells. Further, we highlight recent literature documenting the development of T_VM_ cells as a distinct CD8^+^ T cell lineage as well their biological significance in the context of disease.

## 1. Introduction

Traditionally, CD8^+^ T cells have been considered to exist along a single spectrum; resting naïve CD8^+^ T (T_N_) cells, upon recognition of cognate antigen and subsequent activation, differentiate into effector T cells, which contract upon antigen clearance, leaving a conventional memory T (T_MEM_) cell population. The T_MEM_ cell population is comprised largely of effector memory T cells (T_EM_), found predominantly in tissues and primed for rapid effector function, and central memory T (T_CM_) cells, found mainly in lymph nodes and responsible for self-renewal and supplying the pipeline of effector T cells [1]. T_MEM_ cells are quiescent but poised for activation, and present at a relatively high antigen-specific frequency. These features of CD8^+^ T cell memory underpin their ability to respond rapidly after reencounter with the same antigen and are a hallmark of adaptive immunity. Recently, a novel population(s) of CD8^+^ T cells has been identified, referred to variously as virtual memory T (T_VM_) cells, memory phenotype (MP) T cells, antigen-inexperienced memory T (T_AIM_) cells, and innate memory T (T_IM_) cells, that exhibit many characteristics of T_MEM_ cells—including cell surface phenotype and rapid responsiveness to both antigen-specific and innate stimuli—despite having not previously encountered specific antigen. Herein, we generally refer to this population of antigen-naïve memory phenotype CD8^+^ T cells as T_VM_ cells. The interest in T_VM_ cells stems from their ability to exert robust and rapid effector functions never previously attributed to antigen-inexperienced T cells, their responsiveness to both antigen-specific and innate stimuli, their superior survival capacity and their intrinsic dysfunction in elderly mice and humans [2,3,4,5,6]. The following review highlights how T_VM_ cells have blurred the traditional boundaries between T_N_ cells and T_MEM_ cells and have driven the need for a re-evaluation of conventionally accepted T_MEM_ characteristics. In addition, we discuss recent advances in our understanding of T_VM_ cells, including their development as a distinct cell lineage and their biological relevance in protection from infection and cancers [7,8,9,10,11].

## 2. Conflation of Mouse T_VM_ Cells with Conventional T_MEM_ Cells

In mice, the phenotype of conventional T_MEM_ cell populations is well established [12,13], with all antigen-experienced memory T cells expressing the definitive activation marker, CD44, and differential expression of the lymph node homing receptor, CD62L, allowing distinction between T_CM_ and T_EM_ subsets [14]. However, the discovery of T_VM_ cells has revealed a substantial overlap in the cell surface phenotype of conventional T_CM_ cells and T_VM_ cells in mice, with both expressing high levels of CD44 and CD62L [6,15,16]. Consequently, T_VM_ cells have historically fallen into the conventional phenotypic definition of antigen-experienced CD8^+^ T_CM_ population (Figure 1a). This can be easily overcome by the inclusion of CD49d, an integrin involved in cell trafficking, which is stably upregulated on T_MEM_ cells but, even with advanced age or certain infection models, remains low on T_VM_ cells [3,6]. T_VM_ cells can also be identified by high level expression of IL-2Rβ/IL-15Rβ (CD122) compared to lower expression on T_MEM_ cells, reflecting T_VM_ cell sensitivity to IL-15 [17]. It has recently been demonstrated that conventionally defined T_CM_ cells are, in both young and aged mice, comprised predominantly (~80%) of T_VM_ cells [3]. Even in mice recently infected with LCMV, which induces a robust CD8^+^ T cell memory population [3], over 60% of CD8^+^ CD44^hi^ CD62L^hi^ cells (i.e. conventionally defined T_CM_ cells) are T_VM_ cells [3,18]. Findings regarding the T_CM_ population may therefore be influenced by the inclusion of T_VM_ cells.

Inaccurate attribution of characteristics as a consequence of the conflation of T_VM_ and T_CM_ cells is exemplified by our understanding of the reliance of CD8^+^ T_MEM_ cells on IL-15 for survival [3,6]. The current paradigm indicates that IL-15 is critical for T_MEM_ cell survival. However, recent studies have shown that in young and aged mice lacking IL-15 there is a complete loss of T_VM_ cells (CD44^hi^ CD49d^lo^) whilst T_MEM_ (CD44^hi^ CD49d^hi^) cells are relatively unaffected [3,19,20]. Similarly, in mice lacking CD122, the generation and maintenance of T_MEM_ cells remains relatively intact whilst T_VM_ cells fail to develop [17]. These findings call into question other widely accepted characteristics of T_CM_ cells. Of particular interest is the dependence of T_VM_ and T_CM_ cells on tonic peptide + Major Histocompatibility Complex I molecule (MHCI)-TCR signalling for survival in the periphery. It has long been appreciated that circulating naïve CD8^+^ T cells require low affinity self-pMHCI:TCR interactions in order to provide tonic signals for survival [21], contrasting memory cells whose survival is independent of MHCI but dependent on homeostatic cytokines such as IL-7, IL-15 [22]. However, the requirements for survival of T_VM_ cells is contentious. Early adoptive transfer experiments demonstrated the survival of LCMV-specific memory phenotype (MP) cells in β2m^−/−^ hosts [23]. However, transferred MP cells in this experiment were defined only by high expression of CD44. Expanding on this finding, later adoptive transfer experiments demonstrated that MP cells expressing high levels of CD122 (characteristic of T_VM_ cells) were maintained in the periphery of MHC-Ia^−/−^ mice, whilst MP cells expressing low levels of CD122 failed to survive [24]. These CD122^lo^ MP cells expressed a cell surface phenotype reminiscent of recently activated effector CD8^+^ T cells, including low expression of CD62L. Similarly, inspection of the endogenous population of peripheral CD8^+^ T cells in MHC-I^−/−^ hosts revealed the majority of these cells exhibited CD44^hi^ CD122^hi^ phenotype reminiscent of T_VM_ cells [25]. Thus, the extent to which conventional T_CM_ cells and T_VM_ cells rely on MHC-I for peripheral survival is yet to be definitively determined.

## 3. T_VM_ Cells Are Contained within the Human T_EMRA_ Cell Population

While largely characterized in mice, a putative T_VM_ population has also been identified in humans, which displays both functional and phenotypic similarities to mouse T_VM_ cells [20,26]. These human T_VM_ cells display a differentiated phenotype typically associated with effector memory T cells re-expressing CD45RA (T_EMRA_) (CD45RA^+^CD27^−^), express NK cell receptors (NKRs) such as KIRs and NKG2A, show high expression of Nur77 (indicative of high self-pMHCI affinity) [20], and show high expression of the transcription factor Eomes, with rapid production of IFNγ upon innate-like stimulation [26]. Furthermore, human T_VM_ cells accumulate with age and acquire defects in TCR-mediated proliferation [2,20]. In addition to these parallels with mouse T_VM_ cells, human T_VM_ cells have been detected in human cord blood and thus their development appears to be independent of antigen exposure [26]. The limited study of human T_VM_ cells can be attributed to the lack of definitive surface markers that distinguish them from T_EMRA_ cells (CD8^+^CD45RA^+^CCR7^−^). Currently, identification of human T_VM_ cells is based on the additional expression of NK cell markers, pan-KIR2D and KIR3DL, and/or NKG2A, which separates T_VM_ cells from the entire CD45RA^+^ subset of CD8^+^ T cells. Non-T_VM_ CD45RA^+^ cells are further subdivided into T_N_ cells (CD27^+^CD45RA^+^CD95^−^), T_SCM_ cells (CD27^+^CD45RA^+^CD95^+^) [27] and T_EMRA_ cells. Owing to the overlap in surface marker expression between T_EMRA_ and T_VM_ cells, and the lack of routine inclusion of defining NKRs, T_VM_ cells are typically included within the T_EMRA_ population [28] (Figure 1b).

Despite their apparent similarities, there are key differences that distinguish T_VM_ cells from T_EMRA_ cells. Firstly, T_VM_ cells are considered to be antigen-inexperienced [5,29] whilst T_EMRA_ cells are antigen-experienced memory cells, as evidenced by the observation that they can comprise up to 39% of the CD8^+^ T cells within a given epitope-specific population [30]. Secondly, T_VM_ cells have a higher proliferative capacity than T_EMRA_ cells which are non-proliferative in both young and aged individuals [2]. Thirdly, although a direct comparison between human T_VM_ and T_EMRA_ cell metabolism has not been performed to date, our recent work in mouse models have shown that T_VM_ cells not only have the highest oxygen consumption rate (OCR) of all CD8^+^ subsets in steady state but that it is further increased with infection and ageing [3]. In addition, our study indicates that there is no difference in basal mitochondrial characteristics, such as mitochondrial mass and number of mitochondria per cell, between T_VM_ cells compared to other CD8^+^ subsets [3]. In contrast, T_EMRA_ cells have a lower basal OCR and extracellular acidification rate (ECAR), following overnight CD3 stimulation [31], as well as lower basal mitochondrial mass and fewer mitochondria per cell compared to conventional memory subsets [32].

The inclusion of NKRs to separate putative T_VM_ cells from T_EMRA_ cells in humans marks the beginning of the quest to better investigate this distinct cell population. It is clear that this putative T_VM_ population parallels many of the functional characteristics observed in murine T_VM_ cells, emphasising the need to identify unique and definitive markers for future studies. Indeed, a recent single cell transcriptional analysis of human memory T cells has identified novel subsets of stem-like CD8^+^ memory T cells [27], highlighting the heterogeneity that has confounded a complete understanding of memory phenotype T cells in humans.

## 4. Heterogeneity within the T_VM_ Cell Compartment

Broadly, two populations of antigen-inexperienced MP cells have been described—T_VM_ cells and T_IM_ cells. Whilst T_VM_ cells and T_IM_ cells were originally distinguished from one another by the thymic expression of CD49d, their dependence on IL-15 vs IL-4, and their emergence in the periphery versus the thymus, the two populations are indistinguishable once in the periphery [6,17,33,34]. It is likely that these cells represent the same population, but their original identification in different mouse strains has resulted in the attribution of distinct characteristics. T_IM_ cells are highly abundant in BALB/c mice due to the ability of unconventional PLZF^+^ NKT cells in this strain to produce large amounts of IL-4, facilitating T_IM_ differentiation [33,34,35]. In contrast, T_IM_ cells are not readily detectable in C57Bl/6 mice, however, genetic alterations in these mice, such as knockout of tyrosine kinases ITK and RLK, increases the number of IL-4 producing PLZF^+^ NKT cells and, in turn, increases the number of detectable T_IM_ cells within the thymus and periphery [34,36,37,38,39]. Moreover, while not readily detectable in a WT C57Bl/6 mouse thymus, recent evidence indicates T_VM_ cell differentiation is programmed during thymic development (see below) [40,41,42].

Although peripheral T_VM_ cells in mice are readily identified using cell surface markers such as CD49d, CD44 and CD122, whether or not this population represents a homogenous population of cells or an amalgamation of disparate cell subsets is unclear. Heterogeneity within the T_VM_ population has been suggested by a recent study which used tamoxifen-induced time stamping to analyse T_VM_ cells generated in the neonatal period (day 1) or later in life (day 28) [43]. T_VM_ cells generated early in life (day 1) exhibit a transcriptional profile more akin to a short-lived effector cell (SLEC), as indicated by expression of *tbx21*, *Ifng* and *gzma* genes, compared to those generated later in life (day 28). This heterogeneity translated to differences in functionality, with day 1 T_VM_ cells responding more rapidly to antigen and inflammatory cues linked with increases in effector molecules such as granzyme B and IFNγ. This exaggerated effector response also translated to a greater propensity to adopt a terminally differentiated (KLRG1^hi^ CD62L^lo^) phenotype 41 days post-infection. 

Functional heterogeneity is further observed in a subset of T_VM_ cells that selectively express NK cell markers [20]. NKR expression is a distinguishing feature of mouse T_VM_ cells [6,20,44], and a defining characteristic of human T_VM_ cells. Whilst NKR expression on memory CD8^+^ T cells has conventionally been associated with senescence [45], in T_VM_ cells this subset appears to show heightened functionality, as evidenced by an increased ability to kill MHC-I deficient tumour cells following chemotherapy treatment in both humans and mice [9] (Figure 2).

Adding to evidence of functional heterogeneity within the T_VM_ population, a number of studies have indicated a regulatory role of a subset of CD8^+^CD44^hi^CD122^hi^ cells. Early reports have suggested MP CD8^+^ T cells function similar to that of CD4 T_reg_ cells via IL-10-induced suppression of effector function in activated CD4 and CD8^+^ cells [46] (Figure 2). Later studies revealed that only the PD-1 negative MP subset displayed regulatory functions [47]. In addition, Akane and colleagues characterised these CD8^+^CD44^hi^CD122^hi^ T_reg_ cells and determined they could be further defined from other CD8^+^ MP cells via a lack of CD49d expression, suggesting they were in fact T_VM_ cells [48]. Taken together, these data suggest both phenotypic and functional heterogeneity within the T_VM_ population.

## 5. Heightened TCR Reactivity and Cytokine Sensitivity Are Key T_VM_ Cell Characteristics

TCR reactivity appears to be a key determinant in driving T_VM_ differentiation, phenotype and effector function. Firstly, T_VM_ cells have been shown to express heightened levels of CD5 in mice [49], and Nur77 in humans [20], which are surrogate markers for TCR signal strength [50,51] and thus are indicative of heightened TCR self-reactivity [52]. In addition, TCR repertoire analyses shows a TCR bias in CD8^+^ MP cells [49,53,54], further suggesting the TCR dependence of T_VM_ differentiation. It is likely that this high self-peptide:MHCI reactivity during T_VM_ cell development drives the heightened T_VM_ cell cytokine sensitivity in the periphery, which can, at least in part, be attributed to Eomes expression. Eomes is a Tbox transcription factor which, in CD8^+^ T cells, shows increased expression following activation [55]. In a study by Miller and colleagues, it was shown that Eomes expression could be upregulated during thymic maturation of CD44^hi^CD122^+^ cells, which was attributed to heightened TCR reactivity to self-ligands [53]. Eomes expression has also been shown to bind to the *il2rb* promoter leading to activation and a subsequent increase in CD122 expression [56]. Thus, the heightened self-peptide MHC reactivity of T_VM_ cells appears to upregulate Eomes expression, which in turn leads to increased CD122 expression, driving T_VM_ cell dependence on, and sensitivity to, IL-15. This is also supported by Gett and colleagues who showed that strong TCR engagement, and subsequent signalling, enhanced survival and responsiveness to IL-15, and other cytokines, through increased expression of cytokine receptors [57]. 

Eomes expression in T_VM_ cells has also been shown to be augmented by type I IFN signaling [58]. Indeed, IFNβ signalling resulted in an Eomes-dependent increase of both peripheral T_VM_ cells and thymic T_IM_ cells, and T_VM_ cells were significantly diminished in IFNAR^−/−^ mice [58]. Given the observation that tonic type I IFN signalling is received by SP thymocytes as a normal part of T cell development [59], it seems plausible that type I IFN signalling is essential both in the thymus for T_VM_ lineage differentiation at this SP stage [53], as well as in the periphery for the peripheral maintenance of Eomes expression.

Although a characteristic of memory cell subsets in general, T_VM_ cells are particularly sensitive to a range of homeostatic cytokines, such as IL-12, IL-18, IL-4, and IL-7 [17,20,60,61,62]. As mentioned, their particularly high sensitivity to IL-15 is likely to be due to increased expression of CD122, which increases further with age [3], and leads to a downstream increase in STAT5 phosphorylation following stimulation with IL-15, compared to T_MEM_ cells [2,5,60,63,64,65]. Although the selective impact of cytokines on T_VM_ cells may correspond in part to changes in cytokine receptor expression, age-related changes in T_VM_ cell frequency and function may also be explained by an increase in the levels of these cytokines with aging. For example, there is evidence for elevated IL-15, IL-6, IL-18 and TNF cytokine levels with advanced age, as part of the ‘inflammaging’ process [66,67,68,69]. Outside of IL-15, T_VM_ cells from both mice and humans can be directly activated by other cytokines. Previous in vitro studies of T_VM_ cells have shown that IFNγ production in these cells can be driven by IL-12 and IL-18 stimulation and result in an antigen-independent acquisition of cytotoxic capacity [6,20,60]. 

The role of cytokines in mediating the expansion and effector function of T_VM_ cells is further reinforced in vivo in the context of infection. Baez and colleagues demonstrated that mice infected with *Trypanosoma cruzi* showed enhanced expansion of CD44^hi^ CD8^+^ T cells, owing to increased levels of thymic IL-15 and IL-4, which ultimately promoted antigen-independent proliferation and subsequent protection from parasitemia by this population [70] (Figure 2). Furthermore, the ability of CD44^hi^NKG2D^+^ CD8^+^ T cells to directly kill *Listeria monocytogenes*-infected target cells occurred independently of strong TCR signalling, but was instead NKG2D-dependent and promoted by direct cytokine exposure (IL-12, IL-15 and IL-18) [71] (Figure 2). This innate-like response was required for effective bacterial clearance during the acute stages of infection [71]. In the context of helminth infections, studies have shown that the robust IL-4 production following infection of B6 or BALB/c mice drives antigen-independent T_VM_ cell expansion, which in turn offered significant protection following subsequent viral or bacterial infections, via either innate or antigen-specific mechanisms [7,8,70]. 

Given the heightened cytokine sensitivity of T_VM_ cells, it will be of interest to determine whether changes in the cytokine milieu associated with infections over a life course, or that occur as a natural part of the aging process (‘inflammaging’), are responsible for changes in T_VM_ cell number and function with age. In this way, it seems possible that the same cytokine responsiveness that may impart the rapid responses and semi-differentiated phenotype in T_VM_ cells from young mice and humans, might also be responsible for the acquisition of a senescent phenotype in advanced age.

## 6. NKR Expression as Markers of Functionality and Senescence on T_VM_ Cells 

It has been shown in both mice and humans that a subset of memory phenotype CD8^+^ T cells express NK cell markers, with their frequencies increasing as a result of the ageing process [72,73,74] as well as during infection [75,76]. T_VM_ cells have been reported to express NKRs [20] and appear to demarcate a subset of functionally distinct cells within this population. Following TCR-mediated stimulation, human CD8^+^ T cells expressing NKRs, particular inhibitory NKRs such as NKG2A, KIR2DL and KIR3DL, show reduced effector cytokine production, compared to NKR^−^ CD8^+^ T cells, [77]. Despite this, KIR/NKG2A^+^ CD8^+^ T cells, in contrast to their KIR/NKG2A^−^ counterparts, were able to elicit strong TCR-independent cytotoxic responses via CD16 ligation and enhanced degranulation upon recognition of MHC class I-deficient target cells [26]. Thus, at a functional level, NKR-expressing CD8^+^ T cells appear to display reduced TCR^−^ mediated responses coincident with heightened innate responsiveness.

T cell expression of NKRs is broadly used as an indicator of cellular senescence. As reviewed by Michel and colleagues, the expression of NKRs on CD8^+^ T cells has been linked with classical properties of senescence, including increased expression of cell cycle arrest molecules, such as p53, as well as the markers KLRG1 and CD57, which have been shown to be expressed on terminally differentiated T cells which show poor TCR-dependent signalling activity [78]. Certainly, senescent cells, including senescent T_VM_ cells in aged mice, are characterised both by an upregulation of NKRs and enhanced longevity associated with high Bcl-2 expression [2]. Studies of both mouse and human T cells have shown that expression of inhibitory NKRs results in a decrease in activation-induced cell death in vitro [72,79,80,81]. Moreover, transgenic mice expressing the human KIR2DL3 and HLA-Cw3 ligand showed that ligation of inhibitory KIRs by MHC class I directly resulted in an in vivo accumulation of T_VM_-like cells [72], directly demonstrating a linkage between NKR expression on T cells and enhanced survival, a key characteristic of senescent cells.

NKG2D is another important NKR that can be upregulated on both conventional T_MEM_ cells and T_VM_ cells after activation, and is typically associated with a senescent phenotype when expressed on CD8^+^ T cells [82]. However, despite being a marker of senescence and TCR-mediated dysfunction, the activity of NKG2D on these cells actually facilitates enhanced innate responsiveness. TCR-independent cytotoxicity has been observed following NKG2D engagement on memory phenotype CD8^+^ T cells, through measurement of increased granzyme and perforin production [26,83,84]. In addition, NKG2D has been shown to be expressed on senescent human CD8^+^ T cells and, through its interaction with the ITAM-containing DAP12 protein and Sestrin-2 (SESN2), was able to result in direct activation, cytokine production and cytotoxicity following ligation with its ligand, MICA, or through stimulation with anti-NKG2D antibody [45]. The SESN2-mediated augmentation of NKG2D expression and signalling, along with other NKRs, was concomitant with a downregulation of TCR-mediated signalling. Accordingly, inhibition of SESN2 resulted in a restoration of TCR-mediated signalling and a reduction in NKG2D-mediated, antigen-independent effector function. In mice, this reciprocal TCR-mediated and innate functionality of T cells is exemplified in a study by Anfossi and colleagues who showed that expression of inhibitory Ly49 molecules diminished TCR-mediated T cell activation without loss of sensitivity to IL-15-mediated activation [85]. These data provide a mechanistic explanation for the aforementioned observations of reduced TCR-mediated signalling in T cells expressing NKRs and indicate a reciprocal relationship between innate and adaptive functionality in CD8^+^ MP T cells.

This reciprocal functionality is relevant to the remodelling of function that is observed in both highly differentiated CD8^+^ T_EMRA_ cells in humans (that contain the T_VM_ population) and in T_VM_ cells from aged mice. Namely, the heightened expression of NKRs and sensitivity to cytokines such as IL-15 in advanced age, likely indicates a transition from classical TCR mediated functionality in young individuals, to a primary role in innate effector function, which has been proposed to be beneficial for tumour surveillance and the elimination of senescent cells in tissues [45]. These data highlight the importance of understanding the changing dynamic of T cell stimuli and effector functions over time, and the consideration that the observed ‘senescence’ of CD8^+^ T cells may simply indicate an advantageous functional adaptation in advanced age.

## 7. Transcriptional Regulation of T_VM_ Development and Function

The formation of long-lived CD8^+^ T cell memory after primary responses relies heavily on the binding of early transcription factors (TFs), such as RUNX3, TCF-1 and FOXO1, at pro-memory genes [86,87,88] facilitating chromatin remodelling. These early changes allow memory CD8^+^ T cells to adopt a ‘poised’ chromatin conformation, allowing rapid recall responses via binding of TFs, such as T-bet, at effector gene loci [87]. Similarly, particular TFs have also been implicated in the development or differentiation of T_VM_ cells. As mentioned earlier, the T-box TF Eomes is highly expressed by T_VM_ cells in mice and humans and is critical for T_VM_ formation [6,17]. Importantly these studies demonstrated that early Eomes upregulation in developing CD8^+^ T cells in the thymus prior their adoption of a memory-like (CD44^hi^CD122^+^) phenotype is critical for their development [53]. Other groups have also shown that overexpression of Eomes alone was sufficient to drive antigen inexperienced cells towards a memory phenotype [89]. These studies not only solidify the importance of Eomes in T_VM_ cell development, but also suggest that these cells represent a distinct lineage of CD8^+^ T development (rather than an activation or differentiation stage) that is ratified in the periphery through IL-15 exposure [53]. This substantially changes the paradigm for T_VM_ cell development, which previously considered these cells to arise largely as a consequence of homeostatic proliferation.

Eomes expression likely drives T_VM_ differentiation via its promotion of CD122 expression by binding directly to the *il2rb* promoter [53,56,90]. In addition, it has been shown that Eomes is important for the programming of active enhancers in antigen-naïve MP cells, including T_VM_ cells [89], via binding to RUNX3-bound enhancers, allowing for the chromatin remodelling protein BRG1 to bind and facilitate loop formation. RUNX3 is a critical transcription factor responsible for memory cell reprogramming following TCR stimulation via repression of gene transcription associated with terminal differentiation [88].

T_VM_ development also appears to be controlled by negative transcriptional regulators, such as Bcl11b. Bcl11b is first expressed at the DN2 stage of thymocyte development and its maintenance is required to suppress innate gene expression, including some NKRs and Runx3 [41]. Despite its critical role in regulating positive and negative selection within the thymus [91,92], mice lacking a functional form of Bcll1b accumulated thymocytes and peripheral CD8^+^ T cells with a memory phenotype (CD44^hi^CD122^hi^) that remain functionally active [40,41,42]. These MP cells upregulated transcripts of NKRs as well as increased Eomes expression contributing to their high expression of CD122 [40]. The increase of these cells was found to be due to a cell intrinsic action of Bcl11b and not secondary to elevated thymic IL-4 availability as observed in other genetically modified mouse models [29,34,36,37,38,39,44]. Thus, T_VM_ cell differentiation appears to be tightly controlled through the coordinated action of both positive and negative transcriptional regulation acting in the thymus during T cell development. 

## 8. Epigenetic Regulation of T_VM_ Cells and Age-Related T Cell Dysfunction

The epigenome has been shown to be a critical determinant of CD8^+^ T cell development as well as the ability of CD8^+^ T cells to mount a robust antigen-specific effector response to infections such as Influenza A virus [93,94,95,96,97]. However, little is currently known about the role of epigenetics in the development of T_VM_ CD8^+^ T cells, particularly the development and changing function of T_VM_ cells with age [3,6,18,44]. In the previously mentioned study by Smith and colleagues, it was demonstrated that T_VM_ cell developmental timing determined responsiveness to effector cues and also impacted T_VM_ cell epigenetic identity [43]. The heightened semi-differentiated phenotype seen in day 1 compared to day 28 T_VM_ cells was associated with an increase in chromatin accessibility at gene loci associated with effector function (e.g., *gzma*, *Ifng*, *tbx21*) facilitating rapid transcription. These epigenetic changes resulted in increases in granzyme B and IFNγ production as well as increases in early response to bacterial challenge [43]. Thus, day 1 T_VM_ cells displayed an epigenome pre-programmed towards a SLEC response rather than an MPEC response [98]. These data indicate that T_VM_ functionality, and dysfunction, is likely controlled via underlying epigenetic mechanisms reflected by chromatin accessibility.

Of particular interest is whether the key functional changes associated with T_VM_ cells in aged mice (>18 months), namely the development of cellular senescence and the retention of cytokine responsiveness, are also reflected by epigenetic changes. Further, it is unclear whether the epigenome of aged T_VM_ cells reflects that which is typically observed in conventional memory T cells that have undergone proliferative senescence [99,100]. Broadly, ageing cells tend to exhibit many different epigenetic changes, including the loss of histone proteins, imbalances in post translational modifications, site-specific gains in heterochromatin and aberrant changes in chromatin accessibility [101]. All of these changes result in chromatin instability, transcriptional noise and impaired DNA damage repair all of which contribute to the accumulation of senescent cells as well as the pathology observed with age [102,103]. In an effort to understand the role of epigenetics in the regulation of T-cell specific age-related changes, studies have used ATAC-seq to examined the changes in CD8^+^ T cell chromatin accessibility [104,105]. These experiments demonstrated concentrated changes in chromatin accessibility within aged CD8^+^ T cells, specifically those possessing a memory phenotype. Moskowitz et al demonstrated that CD8^+^ T_N_ and T_CM_ cells adopted a chromatin organisation more akin to differentiated CD8^+^ T cells via their increased accessibility of the bZIP family of transcription proteins, such as BATF, which is critical for effector differentiation [106].

Another significant observation made by these studies was a broad preferential loss in accessibility at promoters rather than enhancers^4^, with affected promoters enriched for NRF1 binding [105] as well as the observed closing at the *il7r* gene promoter [104]. Sites of chromatin closing were also found to be enriched for genes associated with TCR signalling, emphasising the profound effect ageing has on normal T cell function. These studies have provided powerful insight into the molecular mechanisms that underpin human CD8^+^ T cell ageing, but detailed studies on the epigenetics of dysfunctional CD8^+^ T cell subsets, such as T_VM_ cells, are lacking. 

Although clearly important in the differentiation of T_N_ cells to effector T cells and then in the establishment of antigen-specific memory, the role of histone post-translational modifications (PTMs) on T_VM_ cell development and function is poorly studied. However, one key study has revealed an important role for the H3K79 methyltransferase, DOT1L, in T_VM_ cell development [107]. In this study, it was demonstrated that, in the absence of DOT1L from T cells, almost all CD8^+^ T cells exhibited a T_VM_ cell phenotype. Due to the almost complete absence of circulating T_N_ cells in these mice, the authors suggested a role for H3K79me in sustaining the naive state in CD8^+^ T cells, and that deletion of DOT1L may drive cells down a semi-differentiated state independently of antigen exposure [107]. These T_VM_-like cells also appeared during thymic development, suggesting that, similar to Bcl11b, Dot1L is required to suppress development of T cells down a T_VM_ lineage. Further, detailed studies investigating histone PTMs, and epigenetic changes more generally, in the development and maintenance of T_VM_ cells is essential for our understanding of the biology of this population and, by extension and comparison, other CD8^+^ T cell populations.

## 9. Concluding Remarks

As new information emerges it is becoming clearer that memory phenotype T_VM_ cells have a unique function within the immune system, that may evolve quite dramatically over a life course. The identification of this distinct CD8^+^ T cell lineage necessitates a re-evaluation of many of the current paradigms associated with conventional antigen-experienced memory CD8^+^ T cells. Further study should ascertain the heterogeneity of phenotype and function within the T_VM_ population, which should in turn allow for a more precise definition of this population, especially in the human context. Finally, the recent delineation of T_VM_ cells as a distinct developmental pathway, and the identification of several transcriptional and epigenetic controllers of this fate, has highlighted the active mechanisms in place to maintain the naïve CD8^+^ T cell state and prevent excessive T cell activation/innate functionality. Thus, further investigation of T_VM_ populations, while inherently important for our understanding of T_VM_ cell biology, should also serve to advance our understanding of molecular controls that are essential for maintaining adaptive functionality and naïve state of conventional CD8^+^ T cells.

## Figures and Tables

**Figure 1 ijms-21-08626-f001:**
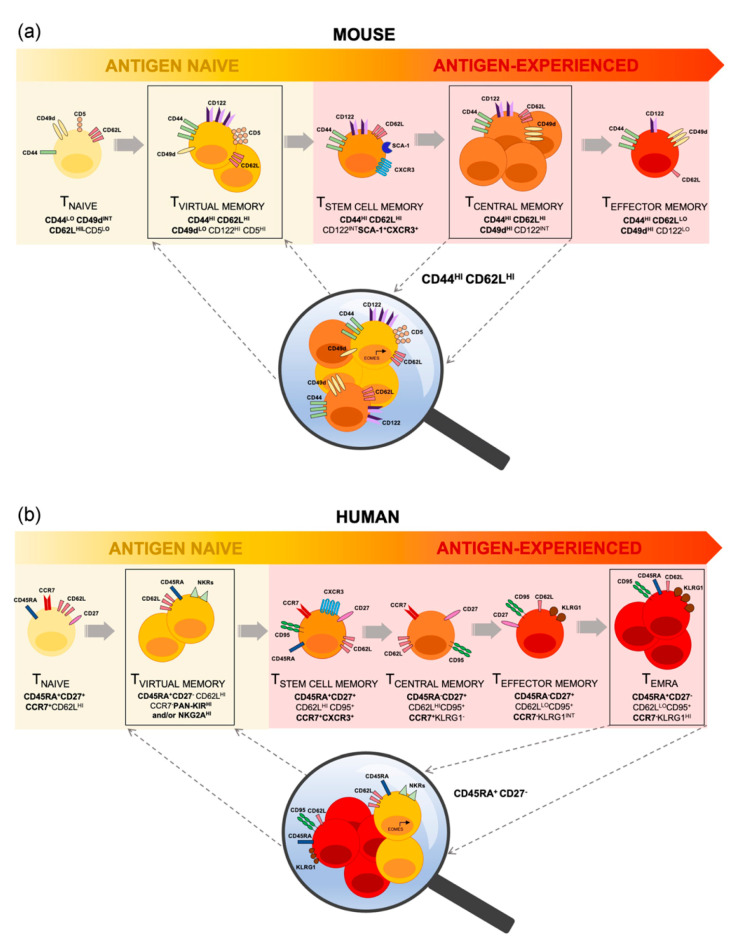
The Differentiation Continuum and Defining Phenotype of Steady-State CD8^+^ T cells. CD8^+^ T_VM_ cells are a semi-differentiated yet antigenically naïve T cell population, indicating the potential for antigen-independent T cell differentiation and representing a link between antigen naïve (yellow shaded) and antigen-experienced (red shaded) memory T cells. T_VM_ cells have historically been phenotypically included within (**a**) the T_CM_ cell population in mice, and (**b**) the T_EMRA_ cell population in humans. Lineage-defining markers (in bold), in conjunction with other additional markers, demarcate T_VM_ cells and reinforce their phenotypic and functional uniqueness.

**Figure 2 ijms-21-08626-f002:**
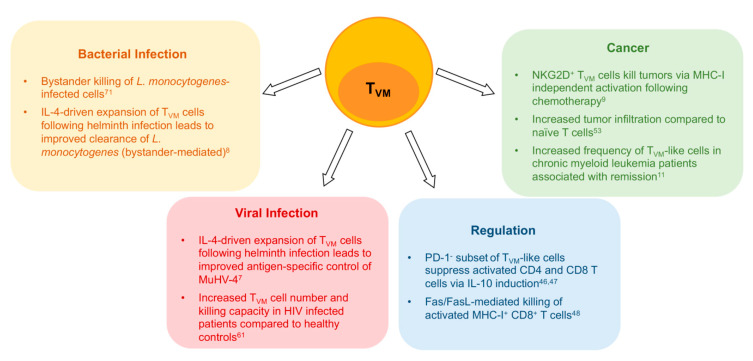
Virtual memory CD8^+^ T cells (T_VM_) participate in various immune responses to pathogens and tumors and may be involved in immune regulation. Boxes indicate the role and possible mechanism of action of T_VM_ cells in different disease or immune contexts.

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
