# Peer review of "Hiding in Plain Sight: Virtually Unrecognizable Memory Phenotype CD8+ T cells"

_ijms, 2020, doi:10.3390/ijms21228626_

Round 1

Reviewer 1 Report

Excellent and timely review on the field from recognized experts in the field.

Only one suggestion: please cite and discuss this very recent paper (it was published when you submitted your rev) 

Two subsets of stem-like CD8+ memory T cell progenitors with distinct fate commitments in humans.

Galletti G, De Simone G, Mazza EMC, Puccio S, Mezzanotte C, Bi TM, Davydov AN, Metsger M, Scamardella E, Alvisi G, De Paoli F, Zanon V, Scarpa A, Camisa B, Colombo FS, Anselmo A, Peano C, Polletti S, Mavilio D, Gattinoni L, Boi SK, Youngblood BA, Jones RE, Baird DM, Gostick E, Llewellyn-Lacey S, Ladell K, Price DA, Chudakov DM, Newell EW, Casucci M, Lugli E.

Nat Immunol. 2020 Oct 12. doi: 10.1038/s41590-020-0791-5. Online ahead of print.

Author Response

We thank Reviewer 1 for their positive feedback and suggestion.

We have cited the suggested paper as follows:-

At the end of the section on TVM cells being contained within the human TEMRA population, we have stated the following (red shows new text):-

"The inclusion of NKRs to separate putative TVM cells from TEMRA cells in humans marks the beginning of the quest to better investigate this distinct cell population. It is clear that this putative TVM population parallels many of the functional characteristics observed in murine TVM cells, emphasising the need to identify unique and definitive markers for future studies. Indeed, a recent single cell transcriptional analysis of human memory T cells has identified novel subsets of stem-like CD8+ memory T cells27, highlighting the heterogeneity that has confounded a complete understanding of memory phenotype T cells in humans."

Reviewer 2 Report

One problem that I see is the author have introduced different receptors, transcription factors and other effectors on different cells and models without providing enough background. In some places this review is very hard to follow. To synthesize the information the authors should make 1-2 figures explaining role of Tvm in different conditions and try to simplify text in some places.

  1. Figure 2 quality is pretty bad and needs to be improved.
  2. “For instance, although TIM cells are barely detectable in the thymus of C57Bl/6 mice, they are highly abundant in BALB/c mice due to the ability of unconventional PLZF+ NKT cells in this strain to produce large amounts of IL-4, facilitating TIM differentiation30–32. A number of genetic alterations on the C57Bl/6 background, including knockout of tyrosine kinases ITK and RLK, increases the number of IL-4 producing PLZF+ NKT cells, thus increasing the amount of detectable TIM cells within the thymus and periphery of C57Bl/6 mice31,33–“  The section here needs to be simplified.
  3. “ater studies revealed that only PD-1- MP subset displayed regulatory functions.” It is not clear if this is PD negative or positive population authors are referring to.
  4. In many places’ authors have used unpublished data to make the point. Unless it is aligned with previously published literature unpublished data should not be part of reviews.
  5. “which has been proposed may be beneficial for tumour surveillance and the elimination of senescent cells in tissues42.” Please improve the sentence.
  6. “and whether the epigenome of aged TVM cells reflects that which is typically observed in conventional memory T cells that have undergone proliferative senescence” Sentence is unclear.
  7. “but detailed studies focusing in particular on the epigenetics of dysfunctional CD8+ T cell subsets, such as TVM cells, are lacking” Please improve the sentence.

Author Response

We thank Reviewer 2 for their suggestions. We have addressed each issue as follows:-

One problem that I see is the author have introduced different receptors, transcription factors and other effectors on different cells and models without providing enough background. In some places this review is very hard to follow. To synthesize the information the authors should make 1-2 figures explaining role of Tvm in different conditions and try to simplify text in some places.

We have now included an additional figure (Figure 2) describing disease or immune conditions in which Tvm cells have been proposed to play a role.

  1. Figure 2 quality is pretty bad and needs to be improved.

We now upload a higher resolution figure

  • “For instance, although TIM cells are barely detectable in the thymus of C57Bl/6 mice, they are highly abundant in BALB/c mice due to the ability of unconventional PLZF+ NKT cells in this strain to produce large amounts of IL-4, facilitating TIM differentiation30–32. A number of genetic alterations on the C57Bl/6 background, including knockout of tyrosine kinases ITK and RLK, increases the number of IL-4 producing PLZF+ NKT cells, thus increasing the amount of detectable TIM cells within the thymus and periphery of C57Bl/6 mice31,33–“  The section here needs to be simplified.

We have altered the sentence structure to try to simplify this concept.

  • “ater studies revealed that only PD-1- MP subset displayed regulatory functions.” It is not clear if this is PD negative or positive population authors are referring to.

We have change this to "PD-1 negative"

  • In many places’ authors have used unpublished data to make the point. Unless it is aligned with previously published literature unpublished data should not be part of reviews.

References to unpublished data have been removed.

  • “which has been proposed may be beneficial for tumour surveillance and the elimination of senescent cells in tissues42.” Please improve the sentence.

We have changed to "which has been proposed to be beneficial for tumour surveillance and the elimination of senescent cells in tissues"

  • “and whether the epigenome of aged TVM cells reflects that which is typically observed in conventional memory T cells that have undergone proliferative senescence” Sentence is unclear.

We have altered the sentence structure here.

  • “but detailed studies focusing in particular on the epigenetics of dysfunctional CD8+ T cell subsets, such as TVM cells, are lacking” Please improve the sentence.

We have altered the sentence structure here.